# Impact of Climate Change and Human Activities to Runoff in the Du River Basin of the Qinling-Daba Mountains, China

Xiaoying Zhang [1,2,†] and Yi He [1,2,*,†] 

1   College of Urban and Environmental Sciences, Northwest University, Xi'an 710127, China; 202121474@stumail.nwu.edu.cn
2   Shaanxi Key Laboratory of Earth Surface System and Environmental Carrying Capacity, Xi'an 710127, China
*   Correspondence: yihe@nwu.edu.cn
†   These authors contributed equally to this work.

**Abstract:** The hydrological response to climate change and human activities plays a pivotal role in the field of water resource management within a given basin. This study was conducted with a primary focus on the Du River basin, aiming to assess and quantify the impacts of climate change and human activities on changes in runoff patterns. The study utilized the Budyko framework in conjunction with the Soil and Water Assessment Tool (SWAT) model to project future changes in runoff while also employing statistical tests like the Pettitt and Mann–Kendall tests to identify abrupt shifts and monotonic trends in the data. The results shows that (1) The analysis of runoff data spanning from 1960 to 2016 revealed a significant declining trend ($p < 0.05$) in annual runoff, with an abrupt change point identified in 1994. The multi-year average runoff depth was determined to be 495 mm. (2) According to the Budyko framework, human activities were found to be the dominant driver behind runoff changes, contributing significantly at 74.42%, with precipitation changes contributing 24.81%. (3) The results obtained through the SWAT model simulation indicate that human activities accounted for 61.76% of the observed runoff changes, whereas climate change played a significant but slightly smaller role, contributing 38.24% to these changes. (4) With constant climate conditions considered, the study predicted that runoff will continue to decrease from 2017 to 2030 due to the influence of ongoing and future human activities. However, this downward trend was found to be statistically insignificant ($p > 0.1$). These findings provide valuable insights into the quantitative contributions of climate change and human activities to runoff changes in the Du River basin. This information is crucial for decision-makers and water resource managers, as it equips them with the necessary knowledge to develop effective and sustainable strategies for water resource management within this basin.

**Keywords:** Budyko framework; climate change; Du River basin; human activities; PLUS model; runoff; SWAT model

## 1. Introduction

Water resources play a pivotal role in sustaining human life and driving socioeconomic development. The effective management and comprehension of water resources are paramount for achieving sustainable development and preserving ecological well-being [1–3]. Surface runoff, as an integral component of water resources, serves as a primary source for human consumption and various societal needs [4–6]. In the study of runoff evolution, it is universally acknowledged that both climate change and human activities contribute significantly to alterations in runoff patterns [7,8]. Climate change directly impacts runoff processes by modifying meteorological elements, including precipitation, temperature, evapotranspiration, insolation, and wind speed. Furthermore, climate-induced adjustments in underlying surface characteristics, such as changes in vegetation cover, exert indirect influences on watershed runoff dynamics [9,10]. These collective climate-related factors shape the quantity and distribution

of runoff within a watershed. Moreover, human activities exert a substantial impact on water resources through changes in land use and cover, as well as the exploitation of water resources. Activities such as urbanization, the construction of reservoirs and dams, hydraulic projects, and artificial water extraction processes are among the human-induced factors that modify runoff patterns in rivers globally [11,12]. These actions can profoundly affect the hydrological processes within watersheds, resulting in shifts in runoff dynamics [13]. Hence, it is imperative to quantitatively assess the driving forces behind runoff changes and enhance our comprehension of the hydrological evolution of watersheds. By conducting a comprehensive evaluation of the contributions of climate change and human activities to runoff variations, it becomes feasible to develop scientifically sound water resource management strategies. Such analyses provide invaluable guidance for effective water resource planning, sustainable utilization, and the mitigation of adverse impacts stemming from runoff changes within watersheds.

There has been a growing interest in the quantitative analysis of hydrological processes influenced by both climate change and human activities in the field of hydrology in recent years [14]. A variety of methods have been developed to assess the drivers of runoff changes in watersheds, including the hydrologic statistical method [15], the elastic coefficient method [16], and the hydrological model method [17]. Among these, the Budyko hypothesis and the hydrological model method are commonly employed [18–20]. The hydrological model method offers a physical understanding of the underlying processes and takes into account the heterogeneity and spatial distribution of watersheds, resulting in a more accurate representation of hydrological dynamics. The Soil and Water Assessment Tool (SWAT) model is widely used both domestically and internationally. Numerous researchers have utilized the SWAT model to investigate runoff changes in various watersheds. For example, Pervez et al. [21] analyzed runoff changes in the Lamaput River, quantitatively assessed the impacts of climate change and human activities on runoff, and validated the suitability of the SWAT model for this specific watershed. Mahaento et al. [22] conducted an attribution study on the Samin Basin in Indonesia, finding that land use changes, particularly related to forests and built-up land, influenced runoff. Fanta et al. [23] employed both the SWAT and the HEC-HMS model to simulate runoff in the Toba watershed, underscoring the significance of land use in runoff simulation. The hydrological model method offers a physical understanding of the underlying processes and takes into account the heterogeneity and spatial distribution of watersheds, resulting in a more accurate representation of hydrological dynamics. Since the SWAT model was introduced in China, it has also been widely applied in numerous basins in China. Yang et al. [24] and Yang et al. [25] studied the effects of climate change, land use, and human activities on runoff changes in the eastern subtropical watershed and the upper Han River watershed in China, respectively. A study conducted in the Wei River Basin by Xu et al. [26] showed that the main cause of runoff reduction was human activities from 1970 to 2017. These methodological approaches and case studies demonstrate the increasing emphasis on understanding and quantifying the drivers of hydrological changes, making strides in addressing the complex interactions between climate, human activities, and runoff patterns.

In contrast to other methods, the Budyko hypothesis presents a notably simple and straightforward approach to computing runoff and evaluating the influence of climate change and human activities. The parameters utilized in the Budyko equation are readily obtainable and hold clear physical interpretations, rendering it more accessible and practicable for implementation compared to intricate hydrological modeling techniques. A noteworthy study by Wang et al. [27] utilized the Budyko hypothesis to examine the interplay between climate and direct human impacts on annual runoff across 413 watersheds in the United States. By deconstructing the Budyko framework, the study quantified the spatial heterogeneity of anthropogenic variability and underscored the combined influence of climate change and human activities on runoff dynamics. Similarly, Xu [28] conducted a comprehensive analysis spanning 296 watersheds in China from 1956 to 2005. Employing the Budyko hypothesis, the study computed and investigated the repercussions of climate change and underlying surface modifications on runoff alterations. The findings illustrated

that climate change predominantly influenced most of the southern watersheds in China, whereas the factors driving runoff changes in the northern watersheds exhibited variability contingent upon their specific geographic locations. These studies showcase the Budyko hypothesis as an accessible and valuable tool for assessing the dynamics of runoff and disentangling the intertwined impacts of climate change and human activities.

The Du River basin, situated within the Qinling-Ba Mountains, holds significant geographical importance. The Qinling-Ba Mountains act as a significant geographic boundary between northern and southern China [29]. The Du River, a first-order tributary of the Hanjiang River, is located entirely within this mountain range, making it a representative transition zone. Moreover, the Du River basin serves as a crucial water source area for the Middle Route Project under the South-to-North Water Diversion. This project is designed to transfer water from the south of China to the water-scarce northern regions. Given this context, the analysis of water resource changes in the Du River basin becomes even more significant. The analysis can provide valuable insights into the availability, distribution, and sustainability of water resources, which are crucial for various sectors, including agriculture, industry, and ecological preservation.

The current research focus on hydrological processes within the Du River basin has predominantly centered on the characterization of hydrometeorological characteristics. This emphasis is evident in studies conducted by Liang et al. [30], Wang et al. [31], Liang et al. [32], and Gu et al. [33]. However, this diversity in research objectives and methodologies has led to variations in research outcomes. To address this inconsistency and provide a more comprehensive understanding of the impact of climate change and human activities on runoff, a synergistic approach combining hydrological methods and model methods is deemed necessary. Furthermore, there has been a limited examination of the effects of anticipated climate change and land use modifications on runoff alterations in the Du River basin. Equally underexplored is the attribution of runoff changes in the watershed. In light of these research gaps, this study aims to bridge this knowledge deficit by leveraging the Budyko hypothesis, the Soil and Water Assessment Tool (SWAT) model, and the PLUS model. Through the application of these analytical tools, this research endeavors to scrutinize the intricate influences of climate change and human activities on runoff dynamics spanning the years 1960 to 2016 within the Du River basin. The primary objectives are to quantify the relative contributions of various factors to runoff changes and to forecast future runoff trends in response to potential land use changes.

## 2. Study Area and Data

### 2.1. Study Area

The Du River basin is situated in the southern region of the upper Hanjiang River. It is recognized as one of the major tributaries of the Hanjiang River and is formed by the confluence of the Sihe River and the Guanduhe River. Geographically, the study area spans Shaanxi and Hubei provinces, specifically located in the Daba Mountains of the Qinling Mountains between 31°21′N and 32°50′N latitude and 109°30′E and 110°40′E longitude (Figure 1). The topography of the watershed is predominantly characterized by subalpine landforms, featuring undulating terrain that gradually decreases in elevation from southwest to northeast [34]. The climate falls within the continental subtropical monsoon climate zone. The distribution of climatic vertical zones varies significantly due to the influence of topography and landforms, resulting in abundant rainfall in the region, which belongs to the humid and watery area in China. The average annual rainfall is as high as 930 mm, with an average annual temperature of 15 °C and an average annual runoff depth of 510 mm. The soil types found in the area primarily consist of yellow-brown loam with a high organic matter content [35]. The main land use patterns are cultivated land, forest, grassland, water area, unused land, and built-up land in the Du River basin. The primary land use patterns observed in the watershed are cultivated land and forest.

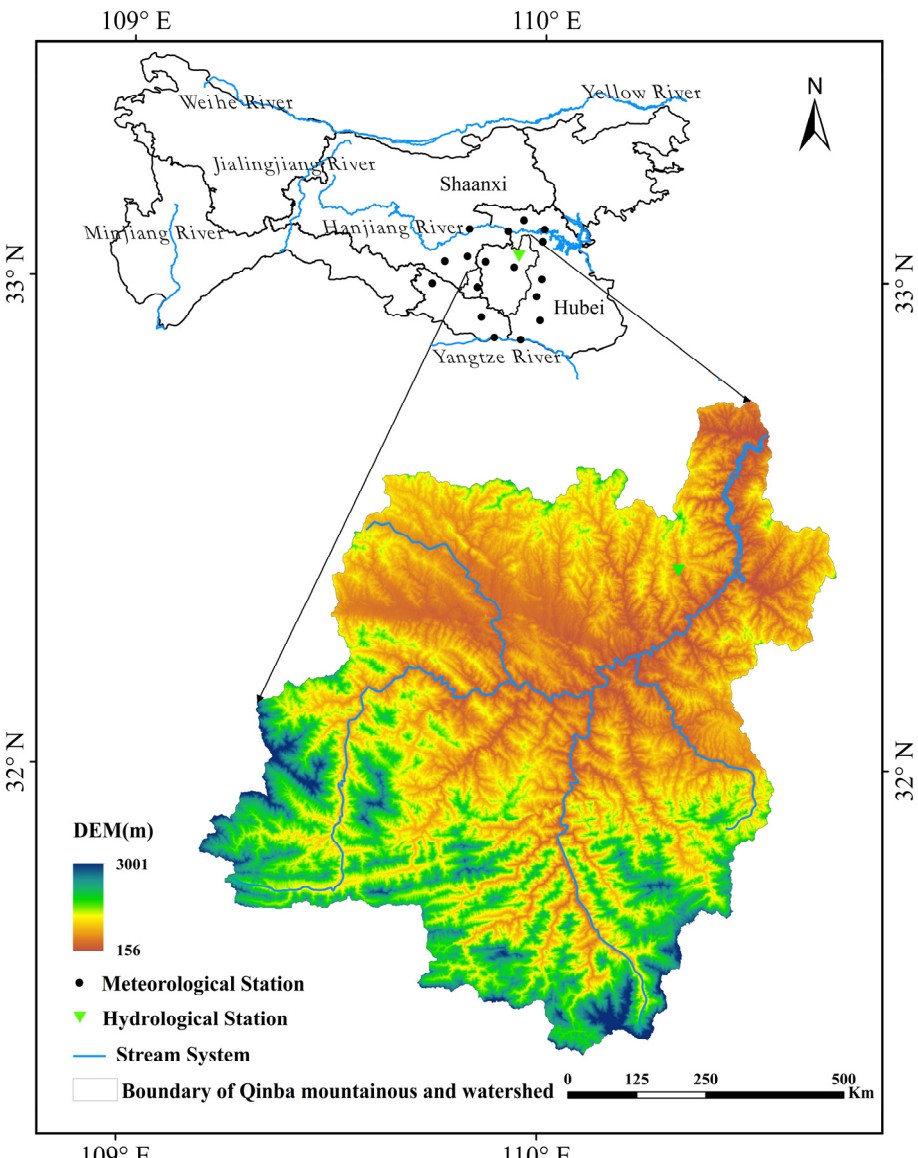

**Figure 1.** Geographic location of the study area and hydrometeorological stations.

The Du River basin is defined based on its boundaries with reference to the Huanglongtan hydrological station. The station is situated in the lower reaches of the Du River, approximately 30 km away from the river's estuary. The Huanglongtan station serves as a key control point for monitoring and assessing the hydrological conditions in the watershed. The watershed area controlled by the Huanglongtan station is significant, covering a total area of 10,995 km$^2$. This area represents approximately 87.9% of the entire Du River watershed. The selection of this station as a reference point allows for a comprehensive analysis of the hydrological processes and runoff characteristics within the major portion of the Du River basin [36]. By focusing on this specific area, the study can provide valuable insights into the hydrological dynamics and water resource management in the larger watershed context.

*2.2. Data*

The researchers collected daily meteorological data from 17 meteorological stations located in and around the Du River basin. These stations were strategically located throughout the watershed to ensure comprehensive coverage of the area (Figure 1). The meteorological data, including mean temperature, precipitation, wind speed, sun duration

in hours, and relative humidity, were acquired from the National Meteorological Information Centre. To calculate the potential evapotranspiration, the researchers employed the Penman–Monteith method [37] based on the above daily scale meteorological data. By calculating the daily potential evapotranspiration of each meteorological station, the annual potential evapotranspiration was summarized. This method is widely used for estimating potential evapotranspiration and takes into account various meteorological parameters [38–40]. The annual temperature, precipitation, and potential evapotranspiration data were then interpolated using the Inverse Distance Weighting (IDW) method.

The annual runoff data for the Huanglongtan hydrological station from 1960 to 2016 were obtained from the Hydrological Information of the Yangtze River, which is published in the Hydrological Yearbook of the People's Republic of China. This dataset provides valuable information on the long-term runoff patterns in the Du River basin. The land use data used were sourced from the Data Center for Resources and Environmental Sciences, Chinese Academy of Sciences. These data cover the period from 1980 to 2015 and provide information on the spatial distribution and changes in land use within the basin. To analyze vegetation dynamics and its influence on runoff, the study utilized monthly 5 km Normalized Difference Vegetation Index (NDVI) data. This dataset, spanning from 1990 to 2016, was obtained from the National Earth System Science Data Center, which is part of the National Science and Technology Infrastructure of China.

## 3. Methodology

In this study, the analysis of runoff changes mainly involves the following steps: (1) trends variation of hydrometeorological elements using MK test and Pettitt test; (2) contribution of runoff changes using Budyko and SWAT model; (3) future runoff changes using SWAT and PLUS model (Figure 2).

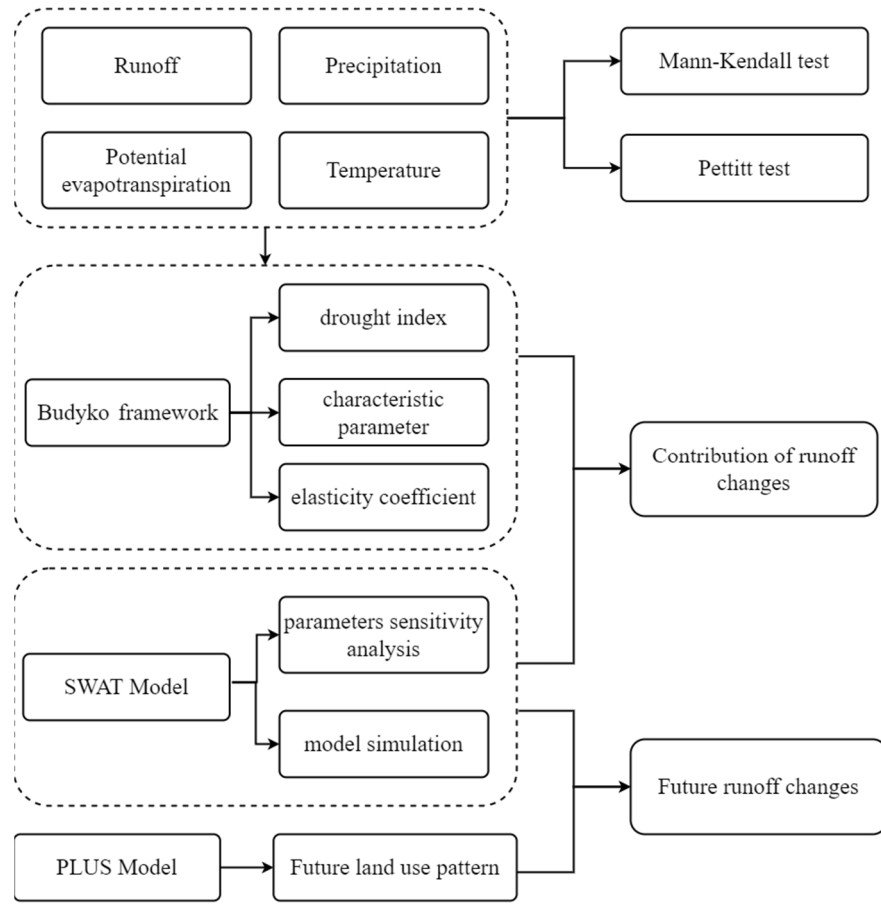

**Figure 2.** Flowchart of the methodology.

### 3.1. Statistics Method

The Mann–Kendall (MK) test [41,42] is indeed a widely used non-parametric statistical test for detecting trends in long time series data [6,43–45]. In the MK test, the null hypothesis is that there is no trend or pattern in the data, and the alternative hypothesis is that there is a monotonic trend present. The test calculates a Z statistic that measures the strength and direction of the trend. If the Z-value is positive, it indicates an increasing trend in the data, whereas a negative Z-value suggests a decreasing trend. The magnitude of the Z-value indicates the significance of the trend. In the MK test, if the absolute value of the Z statistic is greater than a critical value (e.g., 1.65, 1.96, or 2.58), the trend is considered statistically significant at the corresponding significance level (e.g., 0.1, 0.05, or 0.01).

The Pettitt test is indeed a widely applied statistical test in hydrological studies to detect abrupt changes or breakpoints in time series data [46,47]. It is particularly useful for identifying significant shifts in the mean or other characteristics of the data. By applying the Pettitt test to the runoff data in the study area, the researchers aimed to identify and assess any significant abrupt changes or breakpoints in the runoff time series. This information can provide valuable insights into shifts in the hydrological regime and help understand the factors driving these changes.

### 3.2. Budyko Framework

#### 3.2.1. Water Balance Equation

There are many empirical formulas used to describe water balance in watersheds based on the Budyko hypothesis. Choudhury [48] and Yang [49] proposed empirical formulas for calculating annual evapotranspiration and derived the coupled water–thermal balance equations for watersheds on an annual average time scale. It has been widely used in watershed hydrothermal balance calculations and has become a powerful theoretical tool to quantitatively analyze the contribution of runoff changes. The specific expression is described below:

$$Q = P - \frac{PE_0}{\left(P^n - E_0^n\right)^{1/n}} \tag{1}$$

where Q, P, and $E_0$ represent the runoff depth (in mm), precipitation (in mm), and potential evapotranspiration (in mm), respectively. n represents a characteristic parameter reflecting the condition of the underlying surface, which is related to vegetation types, land use, soil characteristics, and topography.

#### 3.2.2. Sensitivity Analysis

Combined with the general equation of hydrothermal coupling assumed by Budyko, the following formulas can be used to obtain the elastic coefficient value of the runoff to potential evapotranspiration, precipitation, and underlying surface [50]. A positive elasticity coefficient indicates that Q increases with an increase in a certain variable, whereas a negative value indicates that Q decreases with an increase in a certain variable:

$$\varepsilon_{E_0} = \frac{1}{(1 + \varphi^n)\left[1 - (1 + \varphi^{-n})^{1/n}\right]} \tag{2}$$

$$\varepsilon_P = \frac{(1 + \varphi^n)^{\frac{1}{n+1}} - \varphi^{n+1}}{(1 + \varphi^n)\left[(1 + \varphi^n)^{1/n} - \varphi\right]} \tag{3}$$

$$\varepsilon_n = \frac{\ln(1 + \varphi^n) + \varphi^n \ln(1 + \varphi^{-n})}{n(1 + \varphi^n)\left[1 - (1 + \varphi^{-n})^{1/n}\right]} \tag{4}$$

where $\phi = E_0/P$ represents the drought index. $\varepsilon_{E_0}$, $\varepsilon_P$, and $\varepsilon_n$ are the sensitivity of runoff to the influencing factors $E_0$, P, and n, respectively. They represent elasticity coefficients.

### 3.2.3. Analysis of Contribution to Runoff Changes

The hydrological period was divided into two time periods according to the abrupt runoff change year. The multi-year average runoff in time period 1 is $T_1$, and the multi-year average runoff in time period 2 is $T_2$. $T_1$ represents the annual mean runoff in the baseline period; $T_2$ represents the annual mean runoff in the evaluation period. The differences of Q, P, $E_0$, and n were calculated for the two time periods. $\Delta Q$, $\Delta P$, $\Delta E_0$, $\Delta n$ are calculated, respectively, as follows (the change between $T_2$ and $T_1$):

$$\Delta Q = Q_{T_2} - Q_{T_1} \tag{5}$$

$$\Delta P = P_{T_2} - P_{T_1} \tag{6}$$

$$\Delta E_0 = E_{0T_2} - E_{0T_1} \tag{7}$$

$$\Delta n = n_{T_2} - n_{T_1} \tag{8}$$

It is assumed that human activities cause underlying surface changes and that this is mainly caused by land use changes. Climate change is represented by precipitation and potential evapotranspiration. The change in runoff caused by a certain factor can be estimated by the product of the change and its partial derivative [51]:

$$dQ = \frac{\partial Q}{\partial P}dP + \frac{\partial Q}{\partial E_0}dP + \frac{\partial Q}{\partial n}dn \tag{9}$$

The above equation can be simplified into the following equations, where $dQ_P$, $dQ_{E_0}$, $dQ_n$ are the runoff variation due to precipitation, potential evapotranspiration, and subsurface changes, respectively:

$$dQ = dQ_P + dQ_{E_0} + dQ_n \tag{10}$$

The contribution of each factor to runoff changes is calculated as follows:

$$C_i = \frac{dQ_i}{dQ} \times 100\% \tag{11}$$

where i represents P, $E_0$, and n, and $C_i$ represents the contribution of P, $E_0$, and n to runoff changes.

### 3.3. SWAT Model

The SWAT model is a widely used distributed hydrological model developed by the Agricultural Research Service of the United States Department of Agriculture (USDA-ARS) [52]. It is designed to simulate and predict various hydrological processes in a watershed. The SWAT model operates at a spatial scale, dividing the watershed into multiple sub-watersheds or hydrological response units (HRUs) based on factors such as topography, land use, and soil characteristics. It incorporates various components to simulate the water balance, including surface runoff, groundwater flow, evapotranspiration, sediment transport, nutrient cycling, and crop growth. SWAT has been widely applied in various regions around the world to study water resource management and land use planning and to carry out environmental assessments [21,22,53]. It provides a suitable tool for evaluating the impacts of different management strategies and policies on the hydrological response of watersheds.

The SUFI-2 (Sequential Uncertainty Fitting version 2) algorithm is a widely used method for calibrating hydrological models, including the SWAT model [54,55]. SUFI-2 is known for its ability to handle parameter uncertainty and provide a range of parameter sets that can reproduce the observed data reasonably well. In this study, the SUFI-2 algorithm was employed to estimate the optimized parameters of the SWAT model for the Du River basin. The algorithm iteratively adjusts the parameters until a satisfactory fit between

the observed and simulated values is achieved. To assess the reliability of the simulation results, parameter sensitivity analysis and model validation were performed.

The Nash–Sutcliffe efficiency coefficient (NSE), correlation coefficient ($R^2$), percent bias (PBIAS), and ratio of RMSE to the standard deviation of the observations (RSR) were used as evaluation criteria for the simulation results [56]. The NSE measures the proportion of the observed data variability that is captured by the model, whereas the $R^2$ represents the degree of linear relationship between observed and simulated values. PBIAS is used to measure the average change tendency of simulated values greater than or less than observed values [57]. RSR changes from the optimal value of 0 to a larger positive value, and a smaller value indicates better model performance [58]. By employing the SUFI-2 algorithm and assessing the simulation results using NSE, $R^2$, PBIAS, and RSR, this study aimed to ensure the reliability and credibility of the calibrated SWAT model for the Du River basin.

The mean observed and simulated runoff values were calculated separately to deduce the contribution in order to quantify the effects of climate change and human activities on runoff changes [59]. The specific calculation equation is as follows:

$$\Delta Q = Q_{r_2} - Q_{s_1} \tag{12}$$

$$\Delta Q_c = Q_{s_2} - Q_{s_1} \tag{13}$$

$$\Delta Q_h = Q_{r_2} - Q_{s_2} \tag{14}$$

$$n_c = \frac{\Delta Q_c}{\Delta Q} \tag{15}$$

$$n_h = \frac{\Delta Q_h}{\Delta Q} \tag{16}$$

where $\Delta Q$ represents the difference between $Q_{r_2}$ and $Q_{s_1}$; $Q_{r_2}$ represents the measured runoff depth after the mutation point that is revealed in the annual runoff depth data by the Pettitt test results; $Q_{s_1}$ represents the simulated runoff depth before the mutation point; $Q_{s_2}$ represents the simulated runoff depth after the mutation point; $\Delta Q_c$ and $\Delta Q_h$ represent the runoff changes caused by climate change and human activities, respectively; and $n_c$ and $n_h$ represent the contribution rates of climate change and human activities to runoff changes, respectively.

### 3.4. PLUS Model

The PLUS model (Patch-generating Land Use Simulation) is a simulation model used to analyze land use changes. It incorporates the Land Expansion Analysis Strategy (LEAS) and the Multiple Random Seeds (CARS) module, enabling the model to simulate the dynamics of land use patches across various land use types.

In this study, 15 influencing land use factors were incorporated, encompassing both natural and economic aspects. To elucidate the relationships between different types of land use expansion and their associated driving factors, the random forest classification algorithm within the Land Use Expansion Assessment System (LEAS) model was employed. This analysis allowed us to quantify the development probability and contribution rate of each land use type. Subsequently, guided by the specific development probability constraints for each land type, land use pattern simulations for the years 2015 and 2025 were conducted using the Cellular Automaton for Rural Systems (CARS) model. In this simulation process, a transition matrix was constructed based on observed land use changes between 2005 and 2015 (Table 1). Furthermore, the neighborhood weight parameters were determined by taking into account the proportion of expansion area for each land type

(Table 2). By employing these methods, the paper aimed to provide a comprehensive understanding of land use dynamics and expansion patterns within the study area.

**Table 1.** Transition matrix.

| Land Use Patterns | Cultivated Land | Forest | Grassland | Water Area | Built-Up Land | Unused Land |
|---|---|---|---|---|---|---|
| Cultivated land | 1 | 1 | 1 | 1 | 1 | 0 |
| Forest | 1 | 1 | 1 | 1 | 1 | 1 |
| Grassland | 1 | 1 | 1 | 1 | 1 | 1 |
| Water area | 1 | 1 | 1 | 1 | 1 | 0 |
| Built-up land | 1 | 1 | 1 | 1 | 1 | 0 |
| Unused land | 0 | 1 | 1 | 0 | 0 | 1 |

**Table 2.** Neighborhood weights.

| Land Use Patterns | Cultivated Land | Forest | Grassland | Water Area | Built-Up Land | Unused Land |
|---|---|---|---|---|---|---|
| Neighborhood weights | 0.2955 | 0.3746 | 0.0728 | 0.1607 | 0.0962 | 0.0003 |

To assess the precision of our model simulation results, two key metrics were utilized: the Kappa coefficient and the Figure of Merit (FOM) coefficient. These coefficients are on a scale from 0 to 1, with higher values signifying superior performance in our simulations [60].

## 4. Results

### 4.1. Trend Variation of Hydrometeorological Elements

Figure 3 illustrates the trends in runoff depth, precipitation, potential evapotranspiration, and temperature within the Du River basin spanning the years 1960 to 2016. To evaluate the statistical significance of these trends and to detect any abrupt changes in runoff depth, the paper employed two commonly used tests: the Mann–Kendall (MK) test and the Pettitt test. In addition, we conducted statistical analysis on each hydrometeorological element to validate the data.

The multi-year average values for runoff depth, precipitation (P), potential evapotranspiration ($E_0$), and temperature (T) in the Du River basin were determined to be 495 mm, 989.1 mm, 898.6 mm, and 14.8 °C, respectively. The results obtained from the Mann–Kendall (MK) test for these variables, as presented in Table 3 and Figure 2, reveal downward trends for all except temperature (T). The MK test produced Z-values of $-2.40$ for runoff depth, $-0.56$ for precipitation (P), $-1.25$ for potential evapotranspiration ($E_0$), and 1.63 for temperature (T). These values imply a decreasing trend in runoff depth, precipitation, and potential evapotranspiration, whereas temperature exhibited an increasing trend within the study area. The corresponding reduction rates were calculated as 3 mm/year for runoff depth, 0.9 mm/year for precipitation, and 0.5 mm/year for potential evapotranspiration. However, the increasing trend of temperature was not significant, and there was almost no change. To ascertain the statistical significance of these trends, *p*-values associated with the MK test were considered. The decreasing trend in runoff depth was found to be statistically significant ($p < 0.05$), indicating a substantial decline in runoff depth over the study period. However, the decreasing trends observed in precipitation, potential evapotranspiration, and temperature were not statistically significant ($p > 0.1$), suggesting that the observed reductions in precipitation, potential evapotranspiration, and temperature were not statistically significant within the study area.

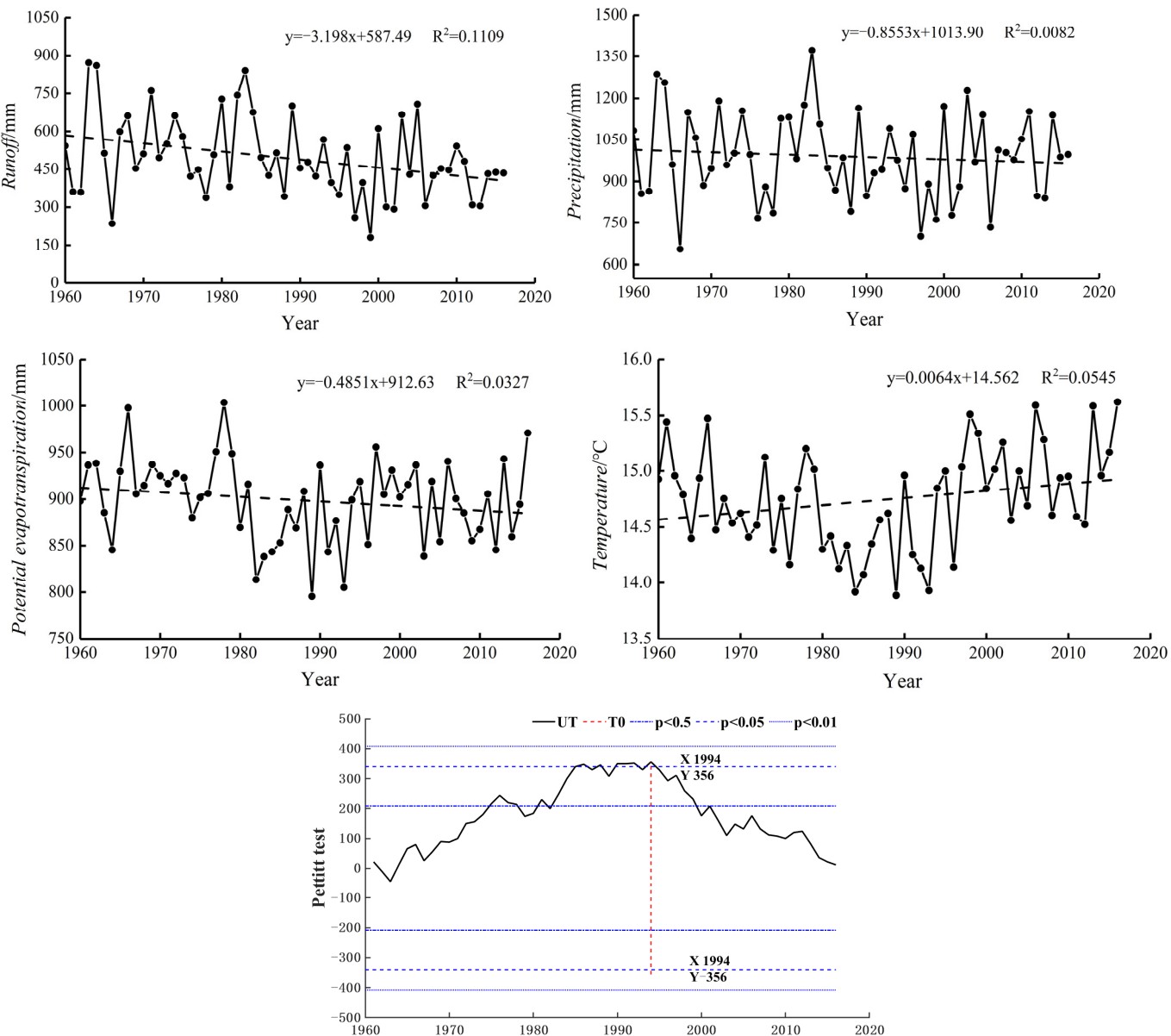

**Figure 3.** Variation tendency and mutation year of hydrometeorological elements in the Du River basin, 1960–2016.

**Table 3.** Trend tests for each hydrometeorological element in the Du River basin, 1960–2016.

| Hydrometeorological Element | Statistic Value | Trend | Significance Level | Amount of Change before and after Mutation |
|---|---|---|---|---|
| Q/mm | −2.40 | Decrease | 0.05 | −117 |
| P/mm | −0.56 | Decrease | - | −40.9 |
| $E_0$/mm | −1.25 | Decrease | - | +1.9 |
| T/°C | 1.63 | Increase | - | +0.4 |

The Pettitt test results revealed a significant mutation in the annual runoff depth data from 1960 to 2016, specifically occurring in 1994 ($p < 0.05$). Consequently, the study period was divided into two periods: the baseline period from 1960 to 1994 and the evaluation period from 1995 to 2016. This division allowed for a comparative analysis of the contributions of different periods. When comparing the evaluation period with the baseline period, it was observed that the annual runoff depth decreased by 117 mm, indicating a substantial reduction in runoff during the evaluation period. Similarly, the

annual P showed a decrease of 40.9 mm during the evaluation period compared to the baseline period. $E_0$ and T exhibited an increase of 1.9 mm and 0.4 °C, respectively, during the evaluation period compared to the baseline period. It might be that the increase in temperature led to an increase in potential evapotranspiration.

Table 4 reveals that $E_0$ exhibited a negative skewness coefficient, indicating left-skewed distribution of the data. On the other hand, the remaining variables exhibited right-skewed distributions. Furthermore, both $E_0$ and T demonstrated smaller coefficients of variation compared to other variables, suggesting more balanced data. The above proves the validity of the data used in this paper.

**Table 4.** Statistical characteristics for each hydrometeorological element in the Du River basin, 1960–2016.

| Hydrometeorological Element | Min | Max | Median | Skewness | Coefficient of Variation |
|---|---|---|---|---|---|
| Q/mm | 180 | 871 | 455 | 0.52 | 0.32 |
| P/mm | 654.4 | 1372.8 | 980.9 | 0.11 | 0.16 |
| $E_0$/mm | 795.5 | 1003.6 | 905.0 | −0.12 | 0.05 |
| T/°C | 13.9 | 15.6 | 14.8 | 0.06 | 0.03 |

*4.2. Contribution of Runoff Changes Based on Budyko Framework*

4.2.1. Runoff Sensitivity Analysis

Table 5 presents the elasticity coefficients of runoff in response to precipitation (P), potential evapotranspiration ($E_0$), and the underlying surface characteristic parameter (n). The drought indices for the baseline and evaluation periods were calculated as 0.92 and 0.96, respectively. The comparison between the two periods indicated an increasing trend in the anthropogenic period, suggesting a certain warm–drying tendency. The elasticity coefficients for P ranged from 1.48 to 1.71, indicating that a 1% increase or decrease in precipitation would result in a corresponding increase or decrease in runoff of 1.48–1.71%. Similarly, the elasticity coefficients for $E_0$ ranged from −0.48 to −0.71, implying that a 1% increase or decrease in potential evapotranspiration would lead to a corresponding decrease or increase in runoff of 0.48–0.71%. Furthermore, the elasticity coefficients for n ranged from −0.63 to −0.72, suggesting that a 1% increase or decrease in the parameter would result in a corresponding decrease or increase in runoff of 0.63–0.72%.

**Table 5.** Statistics in hydrometeorological variables of the Du River basin during 1960–2016.

| Period | $E_0$/P | n | Elasticity Coefficient | | |
|---|---|---|---|---|---|
| | | | $\varepsilon_p$ | $\varepsilon_{E0}$ | $\varepsilon_n$ |
| 1960–1994 | 0.92 | 1.01 | 1.48 | −0.48 | −0.63 |
| 1995–2016 | 0.96 | 1.34 | 1.71 | −0.71 | −0.72 |

In summary, the analysis revealed that runoff was positively correlated with P and negatively correlated with $E_0$ and n. The absolute values of the elasticity coefficients for P were higher than those for $E_0$ and n, indicating that runoff was more sensitive to changes in precipitation compared to potential evapotranspiration and the underlying surface characteristics. When comparing and analyzing the elasticity coefficient values, it was evident that the absolute values for all variables in the evaluation period were greater than those in the baseline period. This suggests that runoff in the watershed was more influenced by both climate change and human activities during the period from 1995 to 2016 and that the sensitivity of runoff to these factors increased over time. Furthermore, the increase in the underlying surface characteristic parameter during the evaluation period indicates significant changes in the underlying surface conditions on account of human activities.

### 4.2.2. Quantitative Contribution of Different Factors to Runoff Changes

Table 6 provides a contribution analysis in the Du River basin. The actual runoff depth decreased by 117 mm, whereas the calculated runoff depth showed a reduction of 129 mm. The small difference between the actual and calculated runoff depths suggests that the method used to evaluate the contribution rate was valid. The contributions of P, $E_0$, and n to the changes in runoff were determined to be 24.81%, 0.77%, and 74.42%, respectively. The variation in the underlying surface, attributed to human activities, had a significant effect on the changes observed in runoff.

**Table 6.** Contribution analysis in the Du River basin.

| Period | $\Delta Q_p$ | $\Delta Q_{E0}$ | $\Delta Q_n$ | $\Delta Q$ | $\Delta A_Q$ | $\delta$ | $C_P$ | $C_{E0}$ | $C_n$ |
|---|---|---|---|---|---|---|---|---|---|
| | mm | mm | mm | mm | mm | mm | % | % | % |
| 1960–1994 1995–2016 | −32 | −1 | −96 | −129 | −117 | 12 | 24.81 | 0.77 | 74.42 |

Note: $\Delta Q_p$, $\Delta Q_{E0}$, and $\Delta Q_n$ represent runoff changes due to $P$, $E_0$, and $n$, respectively; $\Delta Q$ represents calculated runoff depth; $\Delta A_Q$ represents actual runoff depth; $\delta$ represents the difference between $\Delta Q$ and $\Delta A_Q$; $C_P$, $C_{E0}$, and $C_n$ represent the contribution of $P$, $E_0$, and $n$ to runoff changes, respectively.

### 4.3. Contribution of Runoff Changes Based on SWAT Model

### 4.3.1. SWAT Model Construction

The runoff data from the Huanglongtan hydrological station served as the basis for the calibration and validation of the Soil and Water Assessment Tool (SWAT) model within the Du River basin. The development of the SWAT model involved several crucial steps. Initially, a warm-up period was chosen, commencing from 1970, to allow the model to reach a stable state. Subsequently, the calibration phase extended from 1971 to 1975, and the validation phase covered the years 1976 to 1980. Land use data from 1980 were incorporated into the model to ensure an accurate representation of the baseline land use conditions, offering a reliable snapshot of the land use status during the specified baseline period. To account for the intricate hydrological characteristics of the Du River basin, the basin was subdivided into 65 sub-watersheds and 873 hydrological response units. This comprehensive subdivision facilitated a more detailed and precise analysis of the hydrological processes operating within the watershed.

In accordance with the previous analysis, 28 parameters that are directly associated with runoff were initially screened in the SWAT model [61]. To assess their sensitivity, the SUFI-2 algorithm was employed in SWAT-CUP, ultimately identifying 12 parameters that exhibited a T-sensitivity of $\geq |0.95|$ and a P-significant value of $\leq 0.4$. The absolute values of these 12 parameters, in descending order, were as follows: RCHRG_DP.gw, GWQMN.gw, CN2.mgt, EPCO.hru, SOL_K(1).sol, HRU_SLP.hru, SLSUBBSN.hru, GW_REVAP.gw, REVAPMN.gw, OV_N.hru, TIMP.bsn, and SOL_AWC(1).sol.

Figure 4 illustrates the final calibration and validation results based on the monthly runoff data. During the calibration period, the NSE and coefficient of $R^2$ attained values of 0.87 and 0.87, respectively. In the subsequent validation period, NSE and $R^2$ yielded values of 0.78 and 0.75, respectively (refer to Table 7). In addition, PBIAS was within the limits of 11% and RSR was less than or equal to 0.5 during the two periods. These results demonstrate that the simulation outcomes aligned with the required accuracy criteria of the SWAT model. Therefore, the model exhibited excellent performance, indicating its suitability for application in the Du River basin. Furthermore, it can serve as a valuable tool for investigating runoff variations under diverse scenarios in future studies.

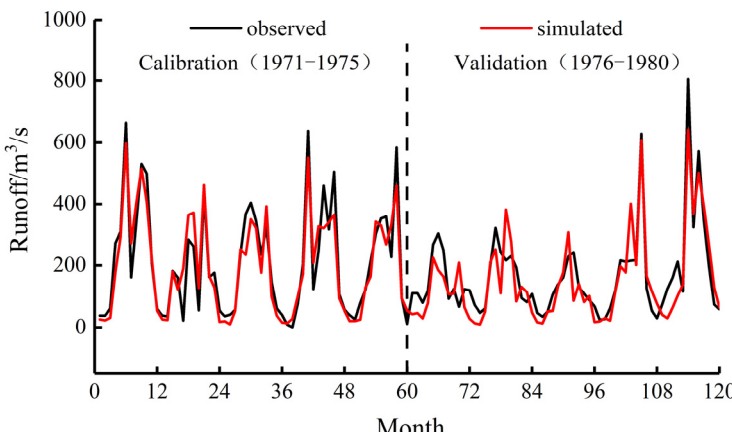

**Figure 4.** Simulation results of the SWAT model in the Du River basin (Note: the left and right sides of the vertical line are calibration and validation period respectively).

**Table 7.** SWAT model simulation effect of runoff.

| Period | $R^2$ | NSE | PBIAS (%) | RSR |
|---|---|---|---|---|
| 1971–1975 (calibration) | 0.87 | 0.87 | 4.20 | 0.36 |
| 1976–1980 (validation) | 0.78 | 0.75 | 10.10 | 0.50 |

### 4.3.2. Quantitative Contribution of Different Factors to Runoff Changes

Table 8 presents the contribution rates obtained through the implementation of the SWAT model. It reveals that climate change contributed to a reduction in runoff of 26 mm, representing a contribution rate of 38.24%. In comparison, human activities resulted in a larger reduction of 43 mm, corresponding to a contribution rate of 61.76%. These findings align with the results obtained through the Budyko framework, indicating that human activities exerted a greater influence on changes in runoff. Consequently, it can be concluded that human activities were the primary driving factor behind the runoff variations observed in the Du River watershed between 1960 and 2016.

**Table 8.** Contribution analysis in the Du River basin.

| Period | $Q_{s_1}$ | $Q_{s_2}$ | $Q_{r_2}$ | $\Delta Q$ | $\Delta Q_c$ | $\Delta Q_h$ | $n_c$ | $n_h$ |
|---|---|---|---|---|---|---|---|---|
| | mm | mm | mm | mm | mm | mm | % | % |
| 1960–1994 1995–2016 | 491 | 465 | 423 | −68 | −26 | −42 | 38.24 | 61.76 |

Note: $Q_{S1}$ represents the simulated runoff depth before the mutation point that was revealed in the annual runoff depth data by the Pettitt test results; $Q_{S2}$ represents the simulated runoff depth after the mutation point; $Qr_2$ represents the measured runoff depth after the mutation point; $\Delta Q$ represents the difference between $Q_{S1}$ and $Q_{r2}$; $\Delta Q_C$ and $\Delta Q_h$ represent the runoff changes caused by climate change and human activities, respectively; $n_c$ and $n_h$ represent the contribution rates of climate change and human activities to runoff changes, respectively.

### 4.4. Prediction of Future Runoff Changes

#### 4.4.1. Prediction of Land Use Change in 2025

The PLUS model was utilized to evaluate the land use expansion strategy for the upcoming decade and predict the land use data for the year 2025. These forecasts were rooted in the observed land use changes that occurred in the Du River basin between 2005 and 2015. The simulation results demonstrated a notably high degree of accuracy, with an overall accuracy rating of 0.95 when comparing the predicted 2015 land use data to the actual data. Furthermore, the Kappa coefficient reached 0.89, signifying a substantial level of agreement, and the Figure of Merit (FOM) coefficient registered at 0.10. These collective results underscore the high level of accuracy and precision exhibited by the PLUS model, rendering it well suited for application within the watershed. Consequently, the model was

deployed to forecast future land use changes, specifically for the year 2025, grounded in the 2015 land use data. These forecasts were employed in subsequent analyses to enhance our understanding of the watershed's dynamic land use patterns.

Based on the information presented in Figure 5, it is evident that the land use pattern within the Du River basin between 2005 and 2015 was characterized by the predominance of cultivated land and forest, with scattered areas of built-up land. In comparison to the land use pattern in 2005, several noteworthy changes were observed. The spatial extent of cultivated land, forest, and grassland decreased by 42.18 km$^2$, 15.79 km$^2$, and 23.44 km$^2$, respectively. Simultaneously, the area occupied by built-up land and water increased by 48.89 km$^2$ and 32.50 km$^2$, respectively. The spatial extent of unused land remained relatively consistent throughout the studied period. Notably, the most substantial change was observed in the water area, which experienced the most significant increase. This was followed by a reduction in the spatial extent of cultivated land, which also exhibited a notable decrease throughout the decade.

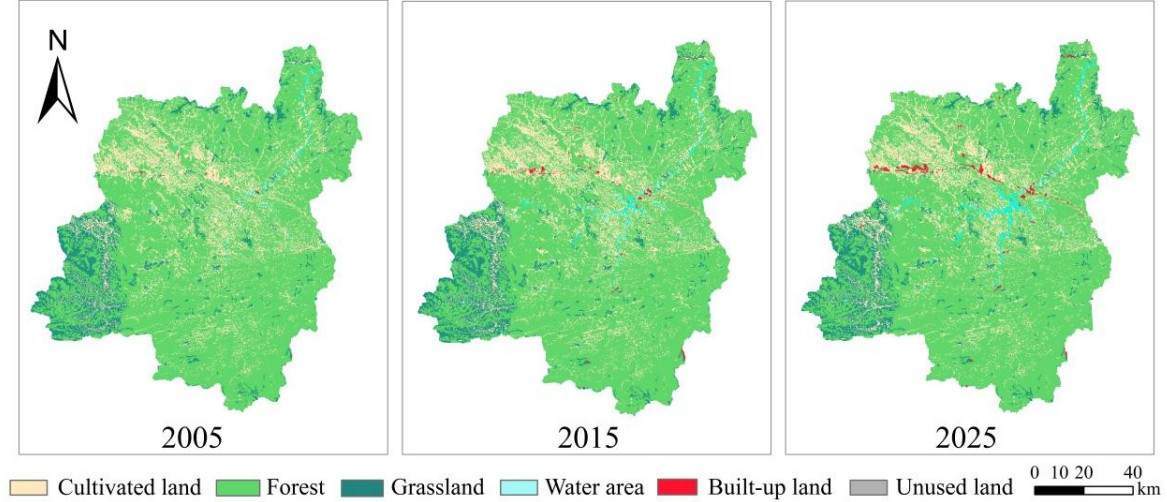

**Figure 5.** Land use changes in the Du River basin for the years 2005, 2015, and 2025.

The future land use pattern is expected to continue the variation tendency observed between 2005 and 2015. Comparing the projected land use pattern with that of 2015, several changes can be anticipated. The area of cultivated land is predicted to decrease by 14.11 km$^2$, whereas the forest area is expected to decrease by 5.37 km$^2$. Similarly, the grassland area is projected to experience a significant reduction of 54.88 km$^2$. On the other hand, the water area and built-up area are anticipated to increase by 44.59 km$^2$ and 29.78 km$^2$, respectively. This may be due to dam constructions increasing the water area. The unused land area is projected to remain relatively stable, with minimal changes. Notably, the most prominent change among all land use categories was the decrease in grassland, followed by the increase in water area.

### 4.4.2. Trend Analysis of Future Runoff Changes

Based on the analysis conducted, it is evident that human activities have had a greater impact on the changes in runoff in the Du River basin compared to climate change in recent years. In light of this, for the purpose of this study, it was assumed that climate conditions would remain unchanged. Additionally, the land use data for the year 2025 were utilized to represent the land use pattern within the watershed between 2017 and 2030. Subsequently, a runoff depth prediction simulation was performed under the condition of land use change. Figure 6 depicts the runoff depth changes projected during the period of 2017–2030, utilizing the SWAT model simulation. The average annual runoff depth in the Du River basin over the next 14 years is estimated to be 510 mm. The MK test was conducted on the annual runoff depth data, yielding a result of −0.11. This result

indicates that the future runoff changes within the study area exhibited an insignificant decreasing trend ($p > 0.1$), with a reduction rate of 2 mm per year. These findings suggest that future human activities will continue to contribute to a gradual decline in runoff within the watershed.

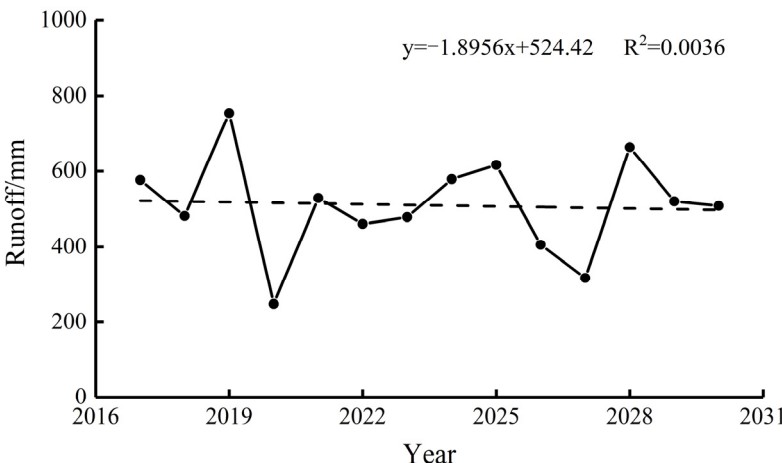

**Figure 6.** Variation tendency of runoff in the Du River basin, 2017–2030.

## 5. Discussion

Climate change and human activities are recognized as the primary drivers influencing studies on runoff changes. The aforementioned study specifically identified human activities as having a more significant impact on runoff changes in the Du River basin compared to climate change. This is in line with the findings of previous studies. In regard to the Du River basin, an important tributary situated in the upper reaches of Hanjiang River, research has indicated the importance of human activities on runoff changes. Niu [62] utilized the SWAT model to simulate runoff in the Du River basin and identified that land use made a significant contribution to runoff during the periods of 1994–1998 and 2005–2009. Peng et al. [63] analyzed the runoff changes across the entire Hanjiang River basin from 1964 to 2015, highlighting vegetation change as the primary factor for the overall decline in runoff within this basin. Xia et al. [64] examined the impacts of climate change and human activities on runoff variations in the upper reaches of the Hanjiang River between 1961 and 2013 using both the elastic coefficient method and the hydrological simulation method, concluding that there was a substantial decrease in runoff during this period, with human activities exerting a greater influence than climate change.

Human activities have the potential to alter hydrological processes through modifications to the underlying surface conditions. These modifications are typically influenced by various factors, such as national policies, hydraulic measures, and land use/cover changes [65]. Among these factors, land use/cover changes exert the most direct and pronounced influence on watersheds in relation to human activities. It is commonly assumed in research studies that topography and soil conditions remain relatively stable over time, allowing for the identification of changes in underlying surface parameters primarily attributed to land use changes or increases in vegetation cover. By understanding the intricate relationship between land use/cover changes and hydrological processes, policymakers and land managers can make informed decisions to mitigate potential impacts on water resources.

Table 9 presents the area and changes in different land use patterns within the Du River basin from 1990 to 2015. The primary land use patterns observed in the watershed were cultivated land and forest. Analysis of the land use data for the four time periods, namely, 1990, 2000, 2010, and 2015, revealed relatively small changes in land use area during each period. In comparison to 1990, the area of cultivated land, forestland, and grassland decreased by 40.73 km², 37.32 km², and 6.05 km², respectively, by the year

2015. Conversely, the water area and built-up land witnessed an increase of 50.72 km² and 33.39 km², respectively. Notably, the area of unused land remained relatively unchanged over the span of 26 years. These findings provide insights into the evolving land use patterns within the Du River basin, with significant changes observed in certain categories whereas others remained relatively stable.

**Table 9.** Area of and changes in land use patterns in the Du River basin, 1990–2015.

| Land Use Patterns | Area (km²) | | | | |
|---|---|---|---|---|---|
| | **1990** | **2000** | **2010** | **2015** | **Change from 1990 to 2015** |
| Cultivated land | 1883.23 | 1889.36 | 1879.17 | 1842.50 | −40.73 |
| Forest | 9058.55 | 9043.23 | 9046.59 | 9021.23 | −37.32 |
| Grassland | 1011.98 | 1020.50 | 1010.06 | 1005.93 | −6.05 |
| Water area | 47.30 | 47.93 | 59.54 | 98.02 | 50.72 |
| Built-up land | 12.31 | 12.39 | 18.12 | 45.70 | 33.39 |
| Unused land | 0.15 | 0.11 | 0.04 | 0.14 | −0.01 |

Table 10 presents the results of the transfer matrix in the Du River basin between 1990 and 2015. The analysis revealed that the conversion between different land use types was relatively small, reflecting the limited changes in land use patterns during this period. Specifically, the increase in the area of water area and built-up land primarily resulted from the conversion of cultivated land and forest. Approximately 22.63 km² of cultivated land and 30.59 km² of forest were converted into watershed, whereas 13.39 km² and 17.49 km² of cultivated land and forest, respectively, were transformed into construction land. Other minor increases in area were predominantly attributed to grassland conversions. These findings highlight the dynamics of land use changes within the Du River basin, with notable shifts observed between specific land use types, particularly cultivated land, forest, and constructed areas, with grassland conversions also contributing to minor changes in the overall land use patterns.

**Table 10.** Transfer matrix of land use patterns in the Du River basin from 1960 to 2015.

| 1990–2015 (km²) | Cultivated Land | Forest | Grassland | Water Area | Built-Up Land | Unused Land | 2015 Total |
|---|---|---|---|---|---|---|---|
| Cultivated land | 1744.85 | 84.43 | 17.93 | 22.63 | 13.39 | | 1883.23 |
| Forest | 78.68 | 8909.59 | 22.09 | 30.59 | 17.49 | 0.11 | 9058.55 |
| Grassland | 16.25 | 23.45 | 965.80 | 3.43 | 3.05 | | 1011.98 |
| Water area | 2.48 | 3.62 | 0.03 | 41.12 | 0.05 | | 47.30 |
| Built-up land | 0.23 | 0.06 | 0.08 | 0.25 | 11.69 | | 12.31 |
| Unused land | 0.01 | 0.08 | | | 0.03 | 0.03 | 0.15 |
| 1990 total | 1842.50 | 9021.23 | 1005.93 | 98.02 | 45.70 | 0.14 | 12,013.52 |

During the period of 1990–2016, there was a notable increase in water area and built-up land within the Du River basin due to human activities. This expansion can be attributed to regional economic development, which had a positive influence, to a certain extent. Previous studies have indicated that, among human activities, the construction of hydraulic facilities is the most direct factor contributing to runoff reduction [66]. Notably, the establishment of the Huanglongtan reservoir in 1978, accompanied by subsequent dam raising and expansion projects within the watershed, has significantly impacted the runoff and sediment volume. Furthermore, the construction of the Pankou hydropower station in the upstream region of the Huanglongtan reservoir in 2011 [36] has enhanced flood control and storage capacity, further influencing the hydrological dynamics within the watershed. In addition to hydraulic developments, the increasing demand for water resources resulting from the expansion of constructed areas has also played a role in altering the runoff dynamics. It is worth mentioning that the influence of The Middle Route Project under the South-To-North Water Diversion, which aims to divert water from the south to

meet the water demand in the north of China, may further contribute to runoff reduction within the watershed.

Furthermore, it is worth noting that the topographic terrain and soil properties within the Du River basin have undergone minimal changes in the short term. As a result, it can be approximated that vegetation change serves as another significant factor contributing to the observed variations in runoff. Vegetation plays a crucial role in modifying hydrological processes by influencing key factors such as interception, infiltration, and evapotranspiration rates. Consequently, alterations in vegetation cover have the potential to significantly impact the quantity and timing of runoff within the watershed. By comprehensively investigating the dynamics of vegetation cover and its interconnectedness with runoff changes, researchers can gain valuable insights into the complex interactions between land use, vegetation patterns, and hydrological processes within the Du River basin.

In this study, the method in [50] was employed to analyze the trend of the Normalized Difference Vegetation Index (NDVI) following a sudden change in runoff. The NDVI, a widely accepted indicator of vegetation growth and coverage, was utilized to investigate vegetation changes within the watershed and assess their impact on runoff dynamics. The study utilized 5 km of NDVI data from 1990 to 2016 to examine vegetation cover changes within the Du River basin. The results indicated that the average annual NDVI for the aforementioned period was 0.748. The Z-statistic value obtained from the Mann–Kendall (MK) test was 5.4202, demonstrating a statistically significant upward trend in the annual average NDVI during the period of abrupt change ($p < 0.01$) (Figure 7).

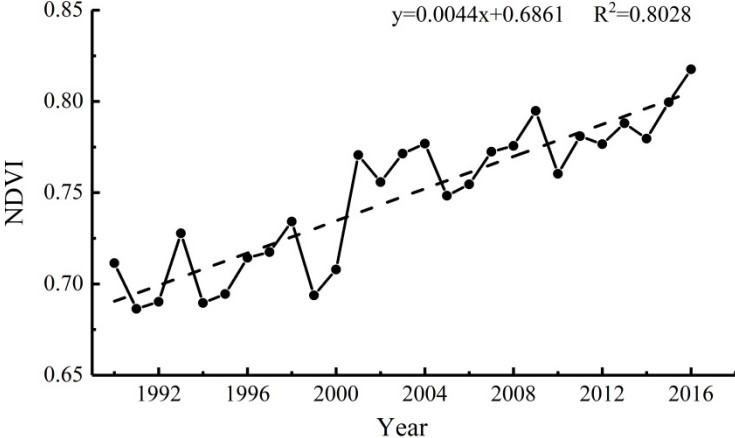

**Figure 7.** NDVI change in the Du River basin during 1990–2016.

Since 1980, a sequence of soil and water conservation measures have been implemented in the Du River basin. Notably, key small watershed management initiatives, such as those undertaken in Zhuxi, Fangxian, and Zhushan city, have played a significant role in mitigating the region's severe soil erosion. These efforts have involved measures such as reforestation, the promotion of economic forestry, slope conversion techniques, and the protection of ditches. These interventions have successfully enhanced vegetation cover and fostered the sustained growth of vegetation [31]. The increase within the Du River basin has had several hydrological implications in vegetation cover. Firstly, it has led to higher interception and evapotranspiration rates within the vegetation canopy [67]. These processes contribute to the overall water storage capacity of the watershed and can modify the production and sink processes within the ecosystem. Consequently, these changes have resulted in a runoff reduction in the Du River basin. Therefore, the expansion of vegetation cover within the watershed can be considered one of the most significant factors contributing to the observed decrease in runoff.

## 6. Conclusions

Based on data from 17 meteorological stations and the Huanglongtan hydrological station, this study employed a comprehensive approach to analyzing runoff changes in the Du River basin. The Budyko framework and the SWAT model were utilized to simulate runoff changes and analyze the evolution characteristics and influencing factors of these changes, with a particular focus on the impact of human activities. Additionally, future runoff trends from 2017 to 2030 were predicted. The main conclusions of this study are as follows:

(1) The annual runoff in the Du River basin showed a noteworthy and statistically significant decreasing trend ($p < 0.05$) from 1960 to 2016, with an abrupt change occurring in 1994. This abrupt change signified a substantial shift in the runoff patterns within the studied period, indicating a significant alteration in the hydrological dynamics of the basin.

(2) The Budyko framework was employed to assess the respective influences of climate change and human activities. The contributions of climate change and human activities to the observed runoff changes were determined to be 25.58% and 74.42%. Additionally, it was observed that runoff changes were more responsive to precipitation variations as opposed to potential evapotranspiration.

(3) Based on the results of the SWAT model, climate change contributed to a reduction in runoff of 26 mm, representing a contribution rate of 38.24%. In comparison, human activities resulted in a larger reduction of 42 mm, corresponding to a contribution rate of 61.76%. Notably, the results indicate that the contribution rate of underlying surface change to runoff depth was the highest.

(4) Under the assumption of unchanged climatic conditions, the analysis of future runoff changes indicated a statistically insignificant decreasing trend ($p > 0.1$), with a reduction rate of 2 mm per year. These findings suggest that, in the absence of significant climate variations, future runoff in the study area will continue to decrease primarily due to ongoing human activities.

**Author Contributions:** Conceptualization, Y.H.; methodology, Y.H.; resources, Y.H.; supervision, Y.H.; validation, Y.H. and X.Z.; visualization, Y.H. and X.Z.; writing—original draft, Y.H. and X.Z.; writing—review and editing, Y.H. and X.Z.; data curation, X.Z.; formal analysis, X.Z.; software, X.Z. All authors have read and agreed to the published version of the manuscript.

**Funding:** This research was jointly supported by the Special Funds of the National Natural Science Foundation of China (grant No. 42341102) and the National Science and Technology Basic Resource Investigation Program (grant No. 2017FY100904).

**Data Availability Statement:** The meteorological data were obtained from the National Meteorological Information Centre (http://data.cma.cn (accessed on 18 May 2021)). The land use data were sourced from the Data Center for Resources and Environmental Sciences, Chinese Academy of Sciences (https://www.resdc.cn (accessed on 18 May 2022)). The Normalized Difference Vegetation Index (NDVI) dataset was obtained from the National Earth System Science Data Center, National Science and Technology Infrastructure of China (http://www.geodata.cn (accessed on 29 May 2022)).

**Conflicts of Interest:** The authors declare that they have no known competing financial interest or personal relationship that could have appeared to influence the work reported in this paper.

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
