# Peer review of "Impact of Climate Change and Human Activities to Runoff in the Du River Basin of the Qinling-Daba Mountains, China"

_remotesensing, doi:10.3390/rs15215178_

Round 1

Reviewer 1 Report

Comments and Suggestions for Authors

The study investigates climatic and anthropogenic activities on streamflow in rivers. Du River Basin from mountainous area, China is selected for the application. Then, some traditional trend methods are applied to streamflow data. The subject is very important and the study is valuable in terms of climate change but the novelty of the study is emphasized insufficiently. Some suggestions and comments to the authors are presented below:

1. A basic flowchart of the suggested methodology should be added in the paper. Thus, the readers can easily follow the application procedures.

2. Some legends on the figures & maps should be presented better and live colours. See Figure 4 …

3. Conclusions part can be improved in the paper. Here is presented in a general concept.

4. What is the novelty of the paper? The used traditional trend methods (as M-K & Pettitt tests) are explained in the paper. Supported and related studies should be strongly presented in the paper by emphasizing the novelty of the paper.

5. Literature part is looking weak. Give new and last updated examples from literature about “trends on hydrometeorological characteristics” as

doi.org/10.3390/f13111776

doi.org/10.54740/ros.2022.016

6. Is the suggested methodology in the paper valid for all areas or is there any limitation or classification for the application?

7. As one important step of the study, the statistical characteristics of used data (e.g. runoff or precipitation data) should be presented in detail. The statistical properties as skewness, coefficient of variation, confidence intervals, distribution characteristics, min, max and median, etc. of used data should be given in a table.

8. The performance metrics part is weak in the paper. More metrics can be calculated to evaluate the application results additionally NSE and R-squared as RMSE, MAE, MSE, RSR (Ratio of RMSE to the standard deviation of the observations) etc. …

9. The resolution of the spatial maps can be increased.

Comments on the Quality of English Language

Check the tenses of the sentences. There are present and past tenses in a paragraph. See the paragraph in the Abstract.

There are some crucial errors.

Keywords should be ordered A to Z.

Use passive sentences. Check the sentences started by “we”. See the line 286 …

Put spaces between numbers and units. See the lines 368, 369 …

Reviewer 2 Report

Comments and Suggestions for Authors

Dear Editor.

I have finished my review on the proposed paper “Impact of climate change and human activities to runoff in the Du River basin of the Qinling-Daba Mountains, China”, remotesensing-2594184-peer-review-v1.

Summary of the manuscript:

In the proposed paper, the authors’ goal is to analyze the quantitative contributions of climate change and human activities to runoff changes. The study predicted future runoff changes and assessed abrupt changes and monotonic trends using the Pettitt and Mann-Kendall tests. The study revealed a significant declining trend (P<0.05) in annual runoff, with an abrupt point occurring in 1994. Annual precipitation and potential evapotranspiration emerged an insignificant decreasing trend.

General review:

1. Generally, the manuscript presents an interesting topic and the specific research seems to include some significant points for the research community of this field.

2. The proposed paper is very well written with very good use of English language. Except some minor grammatical mistakes and word errors. The author should check again the paper to correct these minor mistakes.

3. The proposed paper is very well structured. It begins with the Introduction with some references that helps the reader to get into the subject immediately. In Introduction there is an effort to provide previous studies with similar scientific content, which took place in the research area. Author describes and set very well the scientific problem and how other researchers have approached. At the end of Introduction, authors clearly state the goals of the research. However, I believe that for the specific subject you can enhance the provided literature and the state-of-th-art (see below comments). The literature used is very limited for this kind of paper, only 34 references, and is extremely limited in China region. It is supposed that this study is of international interest and this Journal is also an international journal. Please, add more literature from other countries.

4. The methodology is generally very interesting. However, need some clarifications (see below comments).

5. The results and the discussion are generally OK. However, there some parts that need revisions (see below comments).

Additional points for revision:

In my opinion, the proposed paper could be characterized as a very good research work, complies with the aims of the Remote Sensing Journal. 

Line 34: “… a healthy 33 ecological environment (Chen et al., 2016)”. Please, add more literature.

Line 35-36: “…for human consumption and various societal needs 35 (Chen et al., 2004)”. Please, add more literature and more recent. Here you can add the following studies (doi.org/10.3390/cli11070137 and doi.org/10.1007/s10661-011-2002-1) and other if you want.

Line 36: “…runoff evolution, it is widely recognized that both…”. Please, add literature to support the “widely …”.

Line 38: “…change directly affects runoff processes by altering…”. Please, add literature to support you statement.

Line 40-41: “…such as vegetation cover, have indirect effects on watershed runoff dynamics Please, add literature to support you statement.

Line 43: “…Furthermore, human activities significantly impact water resources….”. Please, add literature to support you statement.

Lines 45-46: “…water extraction processes are among the hu-45 man-induced activities that…”. Please, add literature to support you statement.

Line 55: “There has been increasing interest in quantitatively analyzing…”. Please, add literature to support you statement.

Lines 60-61: “…hydrological model method are commonly used.”. Please, add literature to support you statement.

Lines 64-65: “…Water Assessment Tool (SWAT) model is widely applied both…”. Please, add literature to support you statement.

2.1. Study area: There is no information about the land uses of the watershed.

Lines 146-147: You say that “…17 meteorological stations 146 within the Du River basin”. However, according to figure 1, only 2 meteorological stations are with the watershed. The rest of them are outside. Please, correct accordingly.

Generally in the text: Please, remove the URLs from the text. Add literature and the URLs back in the reference list.

Lines 152-153: “This method is widely used for…”. Please, add literature to support you statement.

Line 170: Please, add literature for the Mann-Kendal method: “Mann, H.B. Nonparametric Tests against Trend. Econometrica 1945, 13, 245–259.”, “Kendall, M.G. Rank Correlation Methods; Griffin: Oxford, UK, 1948.”

Lines 170-171: “…is indeed a widely used non-parametric statistical test for detecting trends…’. Please, add more literature to support you statement.

Line 180: “The Pettitt test is indeed a widely applied statistical test in hydrological studies…”. Please, add more literature to support you statement.

Lines 197-198: “…precipitation (in mm), actual potential evapotranspiration (in mm),..’. Actual or potential???

Line 239: “The SWAT model is a widely used distributed hydrological model…”. Please, add literature to support you statement.

Line 247: “…SWAT has been widely applied in various regions around the world for studying…”. Please, add literature to support you statement.

Lines 251-252: “…algorithm is a widely used method for calibrating hydrological models…”. Please, add literature to support you statement.

Line 256: “…model for the Duhe River basin.”. What is the name of the watershed? Du or Duhe?

Line 259: “The Nash-Sutcliffe efficiency coefficient (NSE)…”. Please, add literature for NSE. (Nash, J., & Sutcliffe, J. V. (1970). River flow forecasting through conceptual models part I—A discussion of principles. Journal of Hydrology, 10(3), 282–290. https://doi.org/10.1016/0022-1694(70)90255-6.

Lines 305-306: “…and potential evaporation 305 in the Du River basin…”. “evaporation” or “evapotranspiration”??

Lines 309-310: “…and potential evaporation 309 (E0) in the Du River…”. evaporation” or “evapotranspiration”?? Check the text for this kind of errors.

Line 310: “…were determined to be 494.75 mm, 989.14 mm, and 898.56 mm, respectively.”. The sum of losses (potential evapotranspiration + runoff) is 1393.31 mm and the precipitation is 989.14. There is a water deficit which is huge (404.17 mm). How this could be reality? Based on these numbers the area can not produce enough water for reservoir storage. Are you sure these numbers are correct?

Lines 322-323: “…specifically occurring in 1994 (p<0.05).”. This maybe could be attributed to the potential increase of temperature. Did you check the trends of temperature???

Lines 330-331: “E0 exhibited an increase of 1.90 mm during…”. This maybe could be attributed to the potential increase of temperature. Did you check the trends of temperature???

Figure 2: The legend of Y axis of “potential evaporation” I think that should be “potential evapotranspiration”.

Lines 342-353: I do not understand why you do not analyze the temperature trends. Temperature is key factor for runoff and evapotranspiration. This is a serious drawback of the study.

Lines 440-442 and 447-449: The increase of water area should be explained here. You should refer here that the dam constructions increased the water area.

Line 501: “…the increase in the area of “watershed” and built-up land…”. Maybe you mean “water” area?

Line 523-524: “…from the expansion of “human-constructed” areas…”. Is there a possibility to exist “non-human” constructed areas, that can affect the runoff? I think that “expansion of constructed areas” is more proper.

Comments on the Quality of English Language

Please, see my comments.

Reviewer 3 Report

Comments and Suggestions for Authors

 Comments and Suggestions for Authors

Impact of climate change and human activities to runoff in the 1 Du River basin of the Qinling-Daba Mountains, China

Manuscript ID: remotesensing-2594184

A brief summary

Water stress resulting from population growth, high water demand, and climate change currently represents an environmentally derived social and economic problem at global, regional, and local scales. The subject of the manuscript fits well in the scope of the Remote Sensing Journal. The importance of this issue becomes obvious from their advantages to contribute with the proactive planning and sustainable management of water resources in the region.

Broad comments

However, for assessment the climate change impact on water balance and hydrological regimes changes in the river basin have to take into the consideration all the water balance components, mainly precipitation, water yield, and evapotranspiration together with its temperature trends. Climate change refers to long-term shifts in temperatures, but it is not discussed in this study. Therefore, I do not recommend publishing it in the present form.

Abstract

This part is too general. It should summarize the major aspects of the entire manuscript. I suggest rewriting it to be more compact.

Introduction

In this part there is an absence of the information for which purposes the model SWAT is mostly used. In addition, the literature review is not up to date, which means the new comparative studies are not mentioned. The hypothesis of this study indicating what is novel and significant in the manuscript are missing. In the motivation of this study aspects and factors, along with causes and consequences, caused by humans in this basin are not analysed and documented.

Line 116:

2. Study area and data

2.1. Study area

In this part the main long-term characteristics have to be described. What is the average annual temperature, precipitation, evaporation, ... for observing period and for long term normal?

Could you, clearly describe all the components for the equations 2 to 11 to be more comprehensive?

Line 309: “The multi-year average runoff depth, precipitation (P), and potential evaporation  (E0) in the Du River basin were determined to be 494.75 mm, 989.14 mm, and 898.56 mm, 310 respectively. The results of the MK test for runoff depth, P, and E0, as presented in Table 3 and Figure 2, indicated downward trends for all three variables.”

This is surprising findings because runoff and potential evaporation should have an adverse trend. Could you explain this?

Line 354: “In summary, the analysis reveals that runoff is positively correlated with P and negatively correlated with E0 and n. The absolute values of the elasticity coefficients for P are higher than those for E0 and n, indicating that runoff is more sensitive to changes in precipitation compared to potential evaporation and the underlying surface characteristics.”

Could you also justify the results and compare them in mm for each of the components of the hydrological balance?

Line 365: Would you be so kind to describe all the components summarised in Table 4?

Reviewer 4 Report

Comments and Suggestions for Authors

See the attachment

Round 2

Reviewer 1 Report

Comments and Suggestions for Authors

I suggest accepting the manuscript. The authors carefully revised the paper by answering each comment from the first round.

Reviewer 2 Report

Comments and Suggestions for Authors

Dear authors.

I studied very careful the revised version provided by the authors. I believe that the authors have significantly improved the paper. They managed to address and respond to all my comments with plausible answers. I have some minor comments to add. A. In line 214 (revised version with red corrections) the "Mann, 1945; Kendall, 1948" literatures must go directly after the phrase of: "The Mann-Kendall (MK) test [Mann, 1945; Kendall, 1948] is indeed a widely used....". B. At the end of this phrase, with the "Zhao et al.,2022; Burgan et al., 2022", please add the following studies (doi.org/10.3390/cli11050106 and doi.org/10.1007/s00704-004-0064-5). C. Furthermore, you have to correct the literature in the text according to Journal's guides. The literature in the text should be with numbers. D. Finally, there are a lot of changes in the text. You have to check again for minor editing mistakes. 

Comments on the Quality of English Language

Minor editing corrections are needed. 

Reviewer 3 Report

Comments and Suggestions for Authors

Comments and Suggestions for Authors: the second time revised

Impact of Climate Change and Human Activities to Runoff in the Du River basin of the Qinling-Daba Mountains, China

Manuscript ID: remotesensing-2594184

I would like to thank you to provide me the opportunity revise this manuscript. The changes those were incorporated had improved its quality considerably. However, there are still several issues that can be resolve.

Abstract

Line 36: “The multi-year average runoff depth was determined to be 494.8 mm.”

The watershed area is 10,995 km², do you think it is important to provide such high resolution? I suggest rounding the numeric values here and values in the whole manuscript.

2. Study area and data

Could you, please, provide the description abut the temperature zone and precipitation distribution in this study?. This description is necessary for this manuscript and can be useful for the comparatives study development.

Figure 1: Would you be so kind to provide more visible mark about the hydrometric station on Du river basin? According to Fig.1 it seems that mark triangle, e.g. Huanglongtan station is not on the main river tributary.

Line 194:To calculate the potential evapotranspiration, the researchers employed the Penman-Monteith method (Allen, 1998).”

Could you, please, specify term the potential evapotranspiration? Is it daily, monthly, or derived from the external values?

Line 327:Qr2 represents the measured runoff after the mutation point; Qs1 represents the simulated runoff before the mutation point…”

Could you provide the information how the mutation point was calculated. It is a very important information, and it is not clear from this description.

Line 385: “…E0 and 0.0064 ℃/year for T.”

This value is not significant, and it can be rounded too.

Table 6: “… Qn represents runoff depth changes…”

This term is not appropriate for the runoff specification. Could you, use the correct term here and everywhere in the text of the manuscript?

Table 8: Would you be so kind to provide the description of each individual parameters those are mentioned in this Table? Note that each Table must be comprehensive without the manuscript text reading.

Discussion

There is still missing information about the comparative studies from this field. Where they performed anywhere in the world and what did they concluded?
